# CT$^2$C-QA: Multimodal Question Answering over Chinese Text, Table and Chart

## ABSTRACT

Multimodal Question Answering (MMQA) is crucial as it enables comprehensive understanding and accurate responses by integrating insights from diverse data representations such as tables, charts, and text. Most existing researches in MMQA only focus on two modalities such as image-text QA, table-text QA and chart-text QA, and there remains a notable scarcity in studies that investigate the joint analysis of text, tables, and charts. In this paper, we present CT$^2$C-QA, a pioneering Chinese reasoning-based QA dataset that includes an extensive collection of text, tables, and charts, meticulously compiled from 200 selectively sourced webpages. Our dataset simulates real webpages and serves as a great test for the capability of the model to analyze and reason with multimodal data, because the answer to a question could appear in various modalities, or even potentially not exist at all. Additionally, we present AED (**A**llocating, **E**xpert and **D**esicion), a multi-agent system implemented through collaborative deployment, information interaction, and collective decision-making among different agents. Specifically, the Assignment Agent is in charge of selecting and activating expert agents, including those proficient in text, tables, and charts. The Decision Agent bears the responsibility of delivering the final verdict, drawing upon the analytical insights provided by these expert agents. We execute a comprehensive analysis, comparing AED with various state-of-the-art models in MMQA, including GPT-4. The experimental outcomes demonstrate that current methodologies, including GPT-4, are yet to meet the benchmarks set by our dataset.

## CCS CONCEPTS

• **Information systems** → **Question answering**.

## KEYWORDS

multimodal Question Answering; Multi-Agent; Multimodal Large Language Model; Text, Table and Chart; Chinese

## 1 INTRODUCTION

Text, tables, and charts are widely used in the fields of finance, healthcare, market research, data analysis, etc., owing to their significant advantages in information presentation: text deepens understanding of topics, providing comprehensive explanations and contextual background; tables present data clearly in a structured format; while charts effectively demonstrate trends and patterns

in data through their intuitiveness. These manifold modalities of data collectively reveal and convey complex information. In the scenario of people browsing this information, the answers to their diversity questions appear in different modalities.

In recent years, there has been a significant interest on Multimodal Question Answering (MMQA), which involves understanding and responding to questions that incorporate multiple modalities, such as text, images, and audio [26]. The initial work on MMQA, as presented in [12], introduced the innovative concept of "Manymodal", which places a spotlight on QA tasks that interact with data spanning more than two modalities. Central to this effort was the development of a diverse dataset, comprised of text, tables, and images, all sourced from Wikipedia. Subsequent research has proposed MMQA datasets with larger scale, more modalities, closer correlation between modalities, and more intricate inference requirements [5, 21, 22, 39, 41]. Although existing MMQA datasets offer significant insights into multimodal interactions, they have overlooked the synergistic potential of combining table, text, and chart data. This trio is fundamental in fields such as statistics and finance, where data interpretation often requires the concurrent analysis of narrative, tabular, and graphical information.

To bridge this research gap, we introduce CT$^2$C-QA, the first **C**hinese reasoning-based QA dataset imitating real webpages, that encompasses **T**ext, **T**ables, and **C**harts, including 9,981 question and answer pairs, and each set of QA pairs associates information about one or more modalities. Our innovative dataset is gathered from 200 websites associated with the National Bureau of Statistics of China[1], encompassing a comprehensive collection of 200 text, 796 tables, and 1051 charts. To mimic the structure of authentic web content, we convert all HTML content into Markdown text format. This involves substituting the HTML content of all tables with specific labels like "table1", "table2", and so on, while ensuring the content of each table is stored separately. Similarly, we represent charts with placeholders such as "img1", "img2", etc., replacing the original hyperlinks found in the HTML source. Additionally, these placeholders are linked to both local storage links and image bed links, providing a versatile and comprehensive representation of the data. This approach maintains a consistent and clear representation of the web content within our framework. Figure1 presents an example and an illustration of the contents encompassed in our dataset: "What was the total profit of the mining industry in January-February 2024?". Answering this question entails (i) understand the content and meaning of the question, (ii) judge the relevance of the question to the data of different modalities: text and table are both relevant to the question, (iii) retrieve information in the relevant modalities, i.e., *"Mining industry", "January-February 2024", "total profits"* all appear together in question, text and table, (iv) integrate the information and generate the answer: *"188.10 billion yuan"*.

---

[1]https://www.stats.gov.cn/

**Query**

2024年1-2月份，采矿业利润总额是多少?

**Translation**： What was the total profit of the mining industry in January-February 2024?

**Text**           **Chart**          **Table**

1—2月份，采矿业实现利润总额1881.0亿元，同比下降21.1%；制造业实现利润总额6134.5亿元，增长17.4%；电力、热力、燃气及水生产和供应业实现利润总额1125.1亿元，增长63.1%。
**Translation**： From January to February, the total profit of the mining industry was 188.10 billion yuan, down 21.1% year-on-year. The total profit of the manufacturing industry was 613.45 billion yuan, up 17.4 percent. The total profit of the electricity, heat, gas and water production and supply industries was 112.51 billion yuan, up 63.1 percent.

采矿业  2024年1-2月份    利润总额（亿元）
mining industry January-February 2024 total profit（billion yuan）
1881
188.10

**Translation**

**Answer**： 1881.0亿元       **Translation**： 188.10 billion yuan

**Figure 1: Example of a CT²C-QA question, answer and context. The distinct keywords in the question are highlighted using various colors. Corresponding information on the webpage is similarly marked with matching colors for easy reference. The answer is specifically indicated with a red font. Each question is associated with a webpage, where the answer might reside in various modal data forms within that page, or it might be that the answer cannot be deduced from the available information. In the example question, the webpage related to the question contains text, three charts and three tables at the same time, and the answer to the question can be found from the text and the table, but there is no relevant information in the statistical graph.**

Our methodology for creating CT²C-QA involves three high-level steps. (a) *Data collection*: We obtain publicly available data from the National Bureau of Statistics of China. Additionally, to preserve the original presentation of the webpage data, we convert the acquired HTML data into Markdown format; (b) *QA pair construction*: Following previous work [49], we generate QA pairs by prompting the Large Language Models (LLMs) to effectively utilize our webpage content; (c) *Quality check*: Based on our sampling inspection findings, we employ varied verification methods for different question-answer pairs. For charts, we designate annotators to manually review every item. In the case of table and text data, we manually inspect a random 25% subset, while entrusting GPT-4 with the evaluation of the remaining 75%.

Currently, researches in MMQA mainly focus on handling two modalities of data. These methods can be broadly categorized into those based on feature fusion, those unified with LLMs, and those employing a divide-and-conquer approach. Although there are also some works dealing with more than two modalities, such as those unified with LLMs [26, 44] and those employing a divide-and-conquer [12, 41] approach, these methods primarily target modalities such as text, tables, and images, without considering chart-type data. Furthermore, while converting all data into text may address some issues, the unique information contained in different modalities cannot be fully described using text alone. The divide-and-conquer approach is employed when the problem is known to occur only in specific categories. It utilizes classification models trained on specific datasets to determine the modality in

which the answer to a given question might appear, based solely on the question. However, this method is not universally applicable because, in new datasets, we often cannot determine the modality in which the answer may appear based solely on a single question.

To tackle this issue, we present AED (**A**llocating, **E**xpert and **D**ecision), a multi-agent system implemented through collaborative deployment, information interaction, and collective decision-making among different agents. Specifically, AED consists of three main components: task allocation, expert processing, and integrated decision-making. The task allocation component integrates all available information to determine in which modalities the answer might appear and provides probabilities accordingly. Experts corresponding to modalities with probabilities exceeding a set threshold are awakened to process the information pertaining to their respective modalities. Finally, the discernment results of all awakened experts are synthesized for integrated decision-making to generate the final answer. AED leverages the strengths of each modality by facilitating seamless communication and cooperation among agents specialized in handling specific data types. This collaborative approach enables AED to effectively integrate diverse modalities, including text, tables, and chart, thus addressing the limitations of existing methods that pay less attention to MMQA containing chart data. Additionally, AED dynamically discriminates the modality of question answers based on data from diverse environments, ensuring its applicability and robustness across various contexts. This final architecture obtains KM = 33.9 and CLKM = 34.3 on our dataset, while the upper-limit human performance is KM = 94.9%,

demonstrating that a substantial amount of future work remains on our new challenge set.

Compared with previous researches, the main contributions of our work are as follows:

- **CT²C-QA**: first Chinese multimodal reasoning-based QA dataset, comprising text, tables, and charts, with a total of 9,981 question-answer pairs. It provides new challenges for existing MMQA methods.
- **AED**: a multi-agent system primarily comprising task allocation, expert processing, and integrated decision-making. It comprehensively analyzes text, table, and chart data, dynamically adapting to various information scenarios.
- Experimental results demonstrate the challenging nature of our dataset and the effectiveness of our method. Our dataset and code will be released later.

## 2 RELATED WORKS

### 2.1 MMQA Datasets

In earlier researches, MMQA datasets primarily focused on two modalities, such as image-text QA [10, 18, 28, 38], table-text QA [6, 16, 34, 52], video-text QA [15, 19, 42, 47], chart-text QA [17, 30, 32]. Each of these datasets presents its unique challenges and has been instrumental in advancing the state of the art in MMQA research. They are commonly used as benchmarks to test the performance of various models in understanding and correlating queries of various modality content with text.

In the real world, scenarios often necessitate the integration and interpretation of information from more than two sources. This necessity led to the development of QA datasets that contain three or even more modalities simultaneously, such as text, images, and tables [12, 22, 41]. However, existing MMQA datasets have not adequately addressed the combination of table, text, and chart data. Recognizing this gap, we introduce the first dataset integrating text, tables, and charts, thereby presenting new challenges to the existing methodologies in the MMQA domain.

### 2.2 MMQA

MMQA datasets present a more comprehensive challenge, requiring models to not only understand and correlate information from multiple sources but also to determine which modality (or combination thereof) is most relevant to answering a given question. We classify the mainstream methods into the following three categories.

**Fusion-based method**, which merges information from diverse modalities into a cohesive representation [9]. Typically, this involves the extraction of features from each modality, followed by their integration using neural networks. Numerous techniques exist for fusing multimodal features, ranging from early [33, 36, 54] to late [1, 4, 11] fusion methods; from Tensor fusion [3, 23, 48, 53] to attention-based fusion [14, 35, 45] approaches. This process adeptly uncovers complex interrelations and synergies between modalities, thereby increasing the accuracy and robustness of the QA system.

**Unified method**, in recent research, various frameworks and models have been proposed to integrate different modality inputs such as images, videos, and audio into the textual feature space of LLMs, enhancing the ability of these models to process and understand multimodal information. For instance, [2, 20] transform

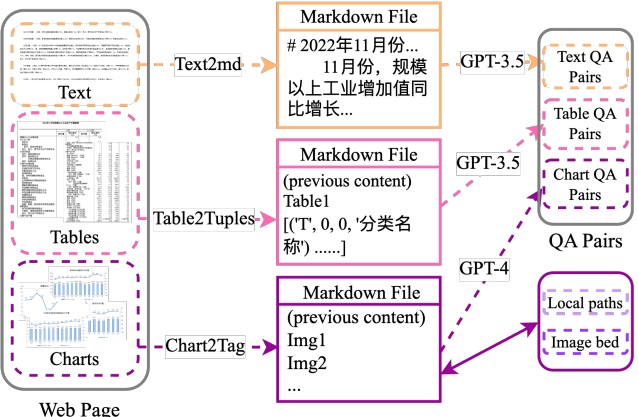

**Figure 2: An illustration of the dataset construction. The orange box represents text data, the pink box contains tables and the purple box contains charts. Following format conversion, these data types are stored within the same Markdown file but in distinct formats. Each chart tag is linked to a local storage path for the corresponding chart and an image bed.**

visual inputs into text, enhancing capabilities in image captioning and visual data interpretation; [27, 56] convert video content into detailed text descriptions, thereby broadening the scope of LLMs in multimedia content analysis and interpretation; [13, 55] demonstrate the translation of audio inputs, including speech, into textual output, facilitating effective interaction with and response to audio-centric content and queries; [25, 40, 58] integrate a diverse range of modalities like text, images, and audio into a unified language model framework, offering a comprehensive and versatile approach to multimodal data processing. However, text alone cannot fully convey the unique information contained in different modalities. Even though LLMs are powerful, they are unable to compensate for all the missing details of the original scene independently.

**Divide and conquer method**, the approach involves training a question classification model on a specific dataset to predict the modality in which the answer to the input question is likely to appear. Subsequently, it selects different models corresponding to the modalities to predict the answer individually [12, 41]. However, this method lacks generalization capability because the classification model is trained on a specific dataset under the assumption that the answers to the questions may correspond to all modalities known. When the answer falls outside the established range, this method becomes ineffective. For example, if the training dataset used for the classification model only contains answers to questions that appear exclusively in text or images, the model will be incapable of handling a question that requires the analysis of both text and image information simultaneously.

In this paper, we integrate the concept of divide and conquer with LLMs to propose AED, a multi-agent system capable of processing text, tables, and charts synthetically, while dynamically adapting to multi-input scenarios.

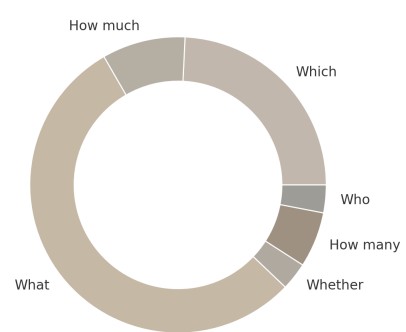
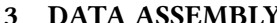

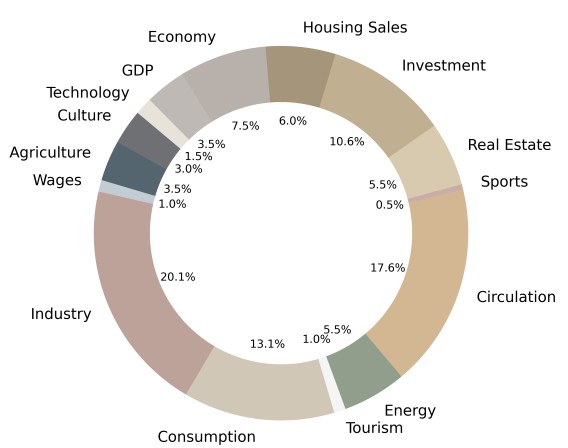

Figure 3: The categories of questions in CT$^2$C-QA for 6 most common first words (statistics after translation).

Figure 4: Distribution of domains in StatChina.

## 3 DATA ASSEMBLY

**Data Collection.** Our data is sourced from the National Bureau of Statistics of China, spanning across more than 1,000 publicly accessible webpages. These pages contain a rich variety of modality data, including text, tables, charts, and more. It is imperative to note that all of this data is publicly available and easily accessible.

**QA Pairs Construction.** The statistical data contains a wealth of information but lacks explicit questions. Therefore, we follow previous works [43, 46, 49] and generate QA pairs automatically. However, due to the unique characteristics of the data, including HTML-formatted tables and charts in image format, as well as redundant HTML tags, we restructure the formats of the various modality data before inputting the original text. To preserve the authenticity of the webpage's format and sequence, the restructuring process, as depicted in Figure 2, entails converting HTML-formatted text into Markdown format, transforming HTML-formatted tables into tuples [57], and substituting instances of statistical charts in the webpage source code with "imgi" tags ( "i" denotes the index of the chart, ranging from 1 to n, where "n" signifies the total number of charts present on the webpage being analyzed). Each tag is linked to a local storage path for the corresponding chart and an image bed. Subsequently, we utilize GPT-3.5-turbo-0125 to generate QA pairs for text and tables, while employing GPT-4-vision-preview to create QA pairs specifically tailored for charts. In particular, to maintain data diversity, when crafting QA pairs, we instruct GPT to generate high-quality pairs that lean towards numerical and entity-based QA pairs, rather than binary yes or no inquiries.

**Quality Check.** After sampling 5% of the data for manual verification, we devise the following procedure to ensure the accuracy of QA pairs for text and tables: a random selection of 25% of the data underwent manual inspection, while the remaining 75% was subjected to verification using GPT-4-0125-preview. Any errors detected are further refined manually. For chart QA pairs, a comprehensive manual inspection approach was employed. Moreover,

during the verification process, we encounter some QA pairs that consistently yielded uniform answers, irrespective of the modality involved. A team of seven graduate researchers in the field of artificial intelligence dedicated a total of 153 hours to manual verification, supplemented by approximately $800 worth of model calls for constructing and verifying QA pairs.

## 4 DATA ANALYSIS

CT$^2$C-QA is composed of data extracted from 200 text, 369 tables, and 494 charts retrieved from 200 webpages. It encompasses a total of 9,981 questions, distributed as follows: 3,335 text-related questions, 3,681 table-related questions, and 1,051 chart-related questions. To highlight the properties of CT$^2$C-QA, we analyze the questions and answers in the question types and answer types. Table 1 shows a comprehensive comparison of related datasets.

**Question Types.** To identify the diversity of the questions, we randomly sample 100 examples from the complete dataset of each modality. Subsequently, these examples are manually categorized. It is noteworthy that some examples are versatile enough to fit into multiple categories. The distribution is illustrated in Figure 3, providing a visual representation of the varied nature of the dataset. Additionally, it should be highlighted that among the questions we randomly select, those pertaining to median difference and contrast analysis comprise 24%, while category analysis and selection accounted for 17%. This underscores the significant demand our dataset places on the model's reasoning capabilities.

**Answer Types.** We employ the keywords extracted from golden answer for the automated categorization of responses. This is executed in a two-step process: Firstly, answers are segregated into numerical and non-numerical based on the presence of numeric elements. Secondly, non-numerical answers undergo further classification into distinct categories such as categories, status, among others. Our dataset encompasses 50.5% of numerical answers and 40.9% of non-numerical. Within the latter category, as detailed in

**Table 1: Dataset statistics and comparison.**

| Dataset | Language | Source | Modality | | | | | | | Question | Word per Question | QA |
|---|---|---|---|---|---|---|---|---|---|---|---|---|
| | | | Text | Table | image | Line | Bar | Chart Pie | Line and Bar | | | |
| ManyModalQA [12] | English | Wikipedia | ✓ | ✓ | ✓ | ✗ | ✗ | ✗ | ✗ | 10,190 | 8.96 | ✓ |
| MMQA [41] | English | Wikipedia | ✓ | ✓ | ✓ | ✗ | ✗ | ✗ | ✗ | 29,918 | 18.2 | ✓ |
| MMCoQA [22] | English | Wikipedia | ✓ | ✓ | ✓ | ✗ | ✗ | ✗ | ✗ | 5,753 | 15.5 | ✓ |
| TTC-QuAli [7] | English | Stat.Canada² | ✓ | ✓ | ✓ | ✓ | ✓ | ✓ | ✓ | - | - | ✗ |
| **CT²C-QA (Ours)** | Chinese | Stat. China | ✓ | ✓ | ✓ | ✓ | ✓ | ✓ | ✓ | 9,981 | 30.2 | ✓ |

**Table 2: All 6 distinct modalities involved, each illustrated with an example and their respective proportions. Common (x,y) means that the answer can be found either in the x mode or the y mode. Note: Overlaps can occur among different modalities. For instance, Q&A for Text in the Common (Text,Table) category exemplifies such intersections. Consequently, the cumulative proportions may exceed 1 due to this potential for modal overlap.**

| Modality | Q&A (Translate) | % |
|---|---|---|
| Text | Q: What is the value added of national culture and related industries in 2021? A: 5,238.5 billion yuan. | 33.4 |
| Table | Q: What is the percentage of the added value of agriculture, forestry, animal husbandry and fishery in the total added value? A: 47.2%. | 36.8 |
| Chart | Q: In the year-on-year growth rate of power generation and average daily production chart, what is the growth rate in November 2022? A: 0.1% | 29.8 |
| Common (Text, Table) | Q: What is the share of the value added of cultural services in the value added of culture and related industries in 2021? A: 64.0% | 18.6 |
| Common (Text, Chart) | Q: How did the volume of imported coal change in November 2022 compared to the previous month? A: Decline. | 5.1 |
| Common (Table, Chart) | Q: In the cement year-on-year growth and average daily production chart, what is the year-on-year growth rate in November 2022? A: -4.7 | 4.2 |

Table 3, we observe distributions such as 23.3% of "Industry Categories", 18.4% of "Statistical Terms", etc.

**Statistics.** We undertake a statistical analysis to delve into the modal composition and the domains encompassed within the dataset. As detailed in Table 2, our dataset comprises a blend of 6 modalities, including "Text", "Table", "Chart", "Text and Table", "Text and Chart" and "Table and Chart". And Figure 4, illustrates our dataset spans a comprehensive range of 15 fields, such as "Industry", "Sports", "Energy" and so on, sourced from statistical reports. This diversity not only show the breadth of our dataset but also highlights its applicability across various domains.

**Table 3: Types of answers in CT²C-QA.**

| Answer Type | % | Example |
|---|---|---|
| Industry Categories | 23.3 | Manufacturing industry |
| Statistical Terms | 18.4 | Growth rate |
| Economic Classification | 15.5 | Average daily production |
| Data Status | 12.6 | Decline |
| Literature | 11.7 | National economic census data |
| Description of Production | 9.7 | 16-25mm |
| Other Categories | 8.7 | Unknown |

## 5  APPROACH

In this section, we propose AED, a multi-agent system comprised of three parts to performance QA on CT²C-QA. The overall framework is illustrated in Figure 5.

### 5.1  Allocating Agent

Our dataset, as depicted in Figure 1, is capable of identifying the webpage relevant to a given question but lacks the precision to pinpoint the specific segment or modality of data associated with it. So we develop an Allocating Agent aimed at discerning the interconnectedness of the question with various data modalities present in a document. The Allocating Agent is structured into

three pivotal modules: the Profile Module, Memory Module, and Action Module.

The Profile Module characterizes the Agent as an adept assistant for multimodal web-based QA. It is tasked with determining the likelihood of answer distribution across different modalities and setting the output format. The Memory Module is bifurcated into two segments: long-term memory, which encompasses the webpage content, and short-term memory, holding the dialogues for each question and answer pair (retaining only the most recent interaction). The Action Module assigns specific probabilities (for instance, $P(text) = a$, $P(table\_i) = b$, $P(chart\_j) = d$, with "i" representing the count of tables on the webpage and "j" indicating the total number of charts) and to activate different Expert Agents based on these

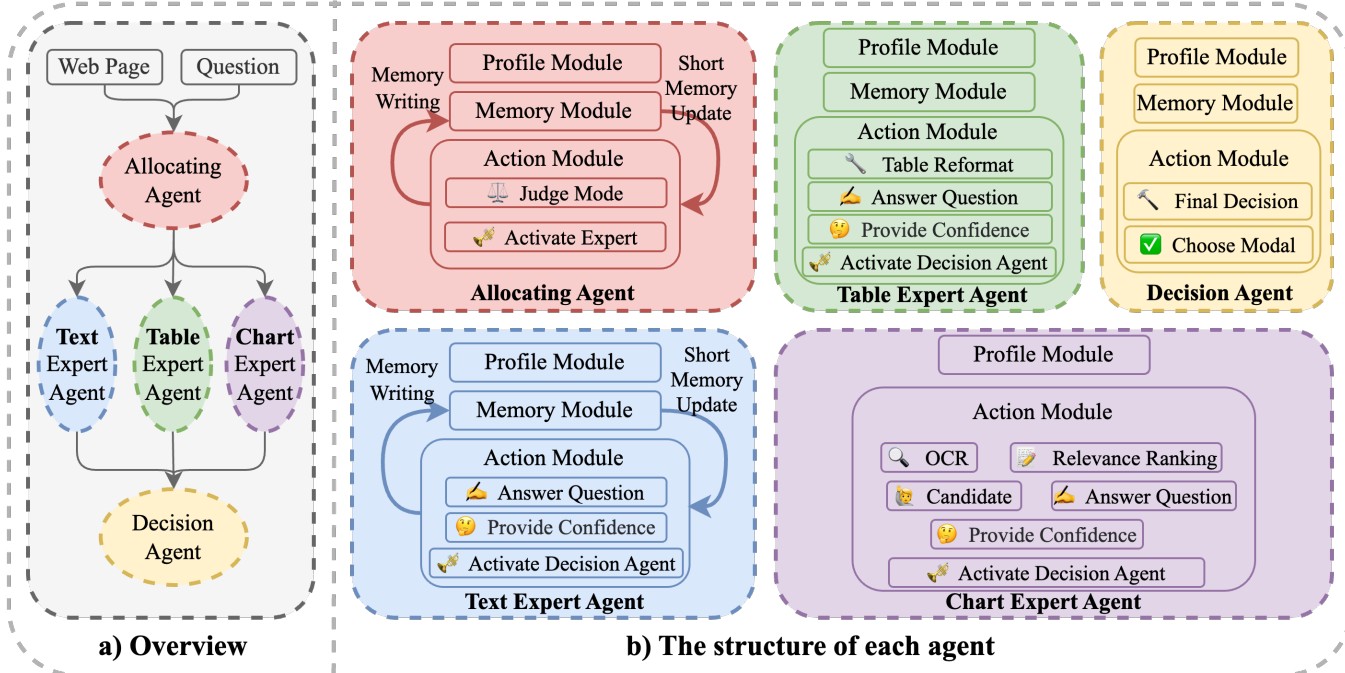

**Figure 5: The overall architecture of AED, which functions by processing both the entirety of webpage content and a question. a) The overview of AED, which displays the interplay and scheduling amongst these various agents. b) The structure of each agent. Different agents within the system are color-coded for clarity: The Allocating Agent is represented in pink. It serves as the initial distributor of tasks and information. The Text Expert Agent, indicated in blue, specializes in handling and interpreting textual content. The Table Expert Agent, shown in green, is focused on processing and understanding table-based information. The Chart Expert Agent, depicted in purple, is adept in analyzing chart data. The Decision Agent, highlighted in yellow, makes final determinations.**

probabilities. In our system, an expert Agent is triggered when the set probability exceeds 0.1. It is important to note that the inputs for the Allocating Agent comprise all web content and the posed questions, wherein tables are represented as tuples and charts are denoted by corresponding tags (Appendix A.1 gives more details).

## 5.2 Expert Agent

We develop three unique Expert Agents, each adept in managing QA tasks specific to different modalities. Mirroring the structure of the Allocating Agent, Text and Table Expert Agents comprise three fundamental modules: the Profile Module, Memory Module, and Action Module. Chart Expert Agent only contains two modules Profile and Memory.

**Text Expert Agent.** The Text Agent receives all text and the question from the webpage as its input. Within its Profile Module, the Agent is designated as a proficient economic analyst, tasked with reading web content and responding to queries, alongside defining the format for the output content. The Memory Module is split into two parts: Long Memory, encompassing the entirety of the webpage's textual content, and Short Memory, which holds the latest round of Q&A. In the Action Module, the Agent is responsible for providing answers as per the requirements, determining the confidence level of each response, and subsequently relaying this

information to the Decision Agent (More details are provided in Appendix A.2).

**Table Expert Agent.** The Allocating Agent assigns probabilities to each specific table, so the input of the Table Expert Agent includes not just the query but also the full text of the table pertinent to the problem. The Profile Module of the Table Expert Agent defines it as a skilled data analyst, acquainting it with the rules for reading tables in tuple format and guiding it to respond to queries in a predetermined format. The Memory Module of this agent consists solely of long-term memory, encompassing the content of each relevant table. In the Action Module, the agent's tasks include converting tables from HTML format to tuple format, answering the question as per the requirements, assessing the confidence level of the response, and forwarding this answer to the Decision Agent. A noteworthy aspect is that the original HTML format of tables often contains extraneous information like tags, while the tuple form simplifies the table's content. Furthermore, we optimize our approach from previous work [57] by eliminating hierarchical representation elements within the tuples, further streamlining the expression (More details are provided in Appendix A.3).

**Chart Expert Agent.** The Chart Expert Agent is an adept statistician designated to handle inquiries related to charts, adhering to a

specific procedural format. It is important to emphasize that the primary objective of our proposed task extends beyond merely answering queries based on a single chart. Instead, it involves the retrieval of the most pertinent chart from a collection of multiple charts prior to providing an answer. Consequently, the principal workflow of our Chart Expert Agent can be outlined as follows: 1) Implementing OCR (Optical Character Recognition) on all charts within an article, this process yields detailed OCR outcomes including the bounding box, numerical values, and their corresponding confidence levels; 2) Extracting and aggregating the values containing Chinese characters from each chart; 3) Independently embedding the aggregated values and the posed question, creating distinct but related data representations; 4) Evaluating the degree of similarity between the embedded chart values and the question, subsequently arranging them in descending order based on similarity scores; 5) Identifying and referencing the chart that exhibits the highest similarity to the question for a precise response. Subsequently, the answer is provided along with an indicated confidence level; 6) Activating the Decision Agent and conveying the gathered information for further action (More details in Appendix A.4).

## 5.3 Decision Agent.

The Decision Agent is composed of three integral parts, each serving a distinct function: 1) Profile Module: This module establishes the Decision Agent as a proficient data synthesis analyst. Its primary role is to analyze the input from all Expert Agents comprehensively. By doing so, it integrates various pieces of information to formulate a final judgment, ensuring a well-rounded and informed decision-making process; 2) Memory Module: This is dedicated to short memory, specifically retaining information from the most recent question-and-answer cycle; 3) Action Module: As the operative heart of the Decision Agent, this module is responsible for delivering the final answer and making necessary selections. It synthetically analyzes the question and the previous inputs and picks the answer of the correct modality as the input. It is noteworthy that our system ultimately outputs both the selected modality and the corresponding answer, enabling a more detailed evaluation of the experimental results and the capability of the Agent (More details are provided in Appendix A.5).

## 6 EXPERIMENT

### 6.1 Setup

We utilize GPT-3.5-turbo-0125, GPT-4-0125-preview, and GPT-4-vision-preview as the foundational models for AED. Specifically, GPT-4-vision-preview is primarily employed for image parsing, while allocation and comprehensive analysis are executed based on GPT-4-0125-preview. All other tasks are completed using GPT-3.5-turbo-0125. In the Action Module of Chart Expert Agent, the OCR task is implemented based on PaddleOCR [8], and the embedding model used in similarity ranking is text-embedding-3-large.

### 6.2 Evaluation Metrics

Prior studies have adopted Exact Match (EM) as the evaluation metric, following the precedent set by [37]. However, EM may not be apt for assessing generative QA tasks. Hence, this paper introduces a novel evaluation method, Keyword Match.

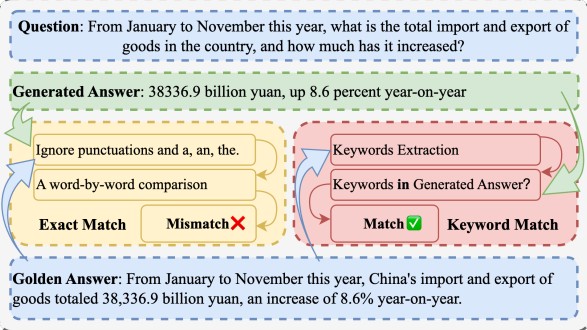

**Figure 6: An illustration of Keyword Match: QA pairs from the Chinese dataset are translated, with the original pairs in a blue box, the generated answer in a green box, the EM metric evaluation in a yellow box, and the KM metric evaluation in a pink box.**

**Keyword Match.** The rise of generative large-scale models has revolutionized sub-tasks within the AI field, prompting research into effective methods for evaluating the generated content. This paper introduces a novel evaluation approach KM to assess accuracy by determining whether the keywords from the golden answer are present in the generated response. As depicted in Figure 6, different from EM, KM extracts the keywords "38,336.9" and "8.6%" from Golden Answer, and ignores the extra symbols when judging "in" with Generated Answer. The final judgment is that "38336.9" and "8.6" are **in** Generated Answer, so the match is successful. Additionally, KM disregards case differences when evaluating words, further showcasing the capabilities of the methods being assessed.

**Cross-Linguistic Keyword Match Validation.** Additionally, since the majority of the models we evaluate are trained using English corpora, they occasionally generate responses in English. To assess the model's comprehension capabilities beyond mere language selection errors, we introduce Cross-Linguistic Keyword Match Validation (CLKM). This method emphasizes the accurate capture of essential information across different linguistic contexts, ensuring a focus on content relevance rather than linguistic form.

**Human Performance.** We assess human performance on a held-out set from the test set containing 300 instances. To evaluate human performance, we present each question alongside its corresponding webpage to three distinct individuals for response. Subsequently, we select the second responses as the human-generated answer and designated the other two as ground truth answers. To assess the accuracy, we calculate the KM and CLKM, comparing the human-predicted answer with the two ground truth answers. The findings revealed that the scores of human performance, indicated by KM = 94.9 and CLKM = 94.9, were significantly superior to those achieved by AED. The primary causes of mismatches can often be attributed to the intricate content of statistical tables and charts, where it is inevitable that the human eye may inaccurately perceive colors and positions. It should be noted that, as the respondents' native language is Chinese, the occasional appearance of English expressions poses no issue, resulting in equal KM and CLKM values.

**Table 4: Performance of various methods and humans. KM stands for Keyword Match and CLKM stands for Cross-Linguistic Keyword Match Validation.**

| Method | Text | | Table | | Chart | | All | |
|---|---|---|---|---|---|---|---|---|
| | KM | CLKM | KM | CLKM | KM | CLKM | KM | CLKM |
| MultiModalQA | 3.2 | 3.9 | 1.7 | 2.1 | 0.9 | 1.0 | 2.0 | 2.4 |
| Human performance | 97 | 97 | 93 | 93 | 95 | 95 | 94.9 | 94.9 |
| **AED (ours)** | 49.2 | 49.6 | 29.6 | 29.7 | 22.1 | 22.7 | 33.9 | 34.3 |

**Table 5: Comparison results of Chart QA on different methods. w/o. rank means that the chart correlation ranking module is not added.**

| Method | KM | CLKM |
|---|---|---|
| MatCha ChartQA | 8.9 | 8.9 |
| MatCha PlotQA-v1 | 2.6 | 2.6 |
| MatCha PlotQA-v2 | 2.7 | 2.9 |
| GPT-4v | 41.0 | 44.7 |
| Llava | 20.5 | 24.1 |
| MiniGPT4-v2 | 12.1 | 12.5 |
| mPLUG-owl1 | 11.6 | 12.5 |
| mPLUG-owl2 | 9.8 | 15.1 |
| **Chart Expert Agent w/o.rank (Ours)** | **49.1** | **54.4** |

## 6.3 Baseline Models

**MMQA.** Given the existing gaps in the fields of text, tables, and charts, we opt to benchmark against the MultimodalQA [41] research, which addresses text, tables, and images. Notably, we are unable to find any open-source code related to Manymodal [12] work for comparison.

**Chart QA.** In the realm of Chart QA, the approach involves training models subsequent to the transformation of charts into tables and the subsequent linearization of these tables. We primarily conduct comparisons with three renowned methodologies: **ChartQA** [29], **PlotQA** [31], and **MatCha** [24]. Furthermore, with the advancements in multimodal large language models, we also choose to include **GPT-4v**[3], **LLaVA-1.6** [25], **MiniGPT4-v2** [58], **mPLUG-owl1** [50], and **mPLUG-owl2** [51] in our comparisons.

## 6.4 Results

We conduct tests across the three modalities—table, text, and chart—and present the evaluation results using the KM and CLKM metrics, as shown in Table 4. Overall, compared to other modals, our method AED soundly outperforms all previous works. The overall KM and CLKM metrics are achieved KM = 33.9 and CLKM = 34.3, respectively, which is a significant leap compared to the KM = 2.0 and CLKM = 2.4 of the the previous method MultiModalQA. However, it still falls short of human performance, which stands at 94.9 for both KM and CLKM. It is noteworthy that in the QA evaluations across

[3]https://platform.openai.com/docs/models/gpt-4-turbo-and-gpt-4

the three modalities, results for the text category are significantly better than those for the table category, which in turn surpasses the chart category. This may be attributed to the Allocating Agent's deeper understanding of text data, followed by tables, and charts being the least comprehensible. This leads to a higher accuracy rate for text-modality related questions. Additionally, within a single webpage, all text data are typically stored in a Markdown file, whereas each table and chart are often stored separately in different files. This means that even after successfully identifying the relevant modality, further identification is required to determine the specific table or chart involved, thereby increasing the potential for errors.

To further illustrate the advantages of the Chart QA task, in our study, we randomly select two questions from each webpage containing a chart and documented their URLs for testing purposes. It is noteworthy that current multimodal large-scale models, as well as Chart QA models, are limited to processing only one chart at a time. To ensure fairness and better demonstrate the varying capacities of different methods in understanding charts, we specifically chose the Chart Expert Agent from AED for our QA task. Additionally, we omit the ranking module to simplify the task into a direct question-and-answer format focused on a single chart. As shown in Table 5, models that are trained and fine-tuned using previous Chart QA datasets have demonstrated suboptimal performance. In contrast, general-purpose multimodal large language models, such as GPT-4v and Llava, have exceeded expectations in the Chart QA task.Compared with these works, our Chart Expert Agent has an absolute improvement of KM = 49.1 and CLKM = 54.4 under the same conditions. Additionally, it has been observed that the performance of most methods improves under the CLKM metric. This improvement is attributed to the focus shifting away from language consistency towards the models' ability to parse and reason about the data presented in charts.

## 6.5 Analysis

The results show that our AED for CT$^2$C-QA effectively outperforms the previous method and shows remarkable results in the task with only a single modal QA of chart. However, compared with the excellent single-modal QA of chart, the overall AED method is still unsatisfactory. We consider that is due to the serial operation of the AED method, which progresses from the Allocating Agent to the Expert Agent, and finally to the Decision Agent. When the Allocating Agent makes an error in modality classification, the likelihood of correctly selecting the appropriate object from multiple tables and charts is consequently reduced. This, in turn, leads to a cumulative increase in errors at each subsequent stage.

## 7 CONCLUSION

We introduce CT$^2$C-QA, a new Chinese multimodal QA dataset comprising 9,981 QA pairs across text, tables, and charts, presenting fresh challenges to MMQA research. We also develop a multi-agent system AED for unified reasoning across these modalities. To better evaluate parsing and reasoning capabilities, we introduce new metrics, KM and CLKM. Despite our advances, human performance still significantly outstrips our methods, highlighting extensive opportunities for further exploration in this field.

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

# A APPENDIX

## A.1 Details of Allocating Agent

## A.2 Details of Text Expert Agent

## A.3 Details of Table Expert Agent

## A.4 Details of Chart Expert Agent

## A.5 Details of Decision Agent

Received 20 February 2007; revised 12 March 2009; accepted 5 June 2009

