# OpenReview forum: "CT2C-QA: Multimodal Question Answering over Chinese Text, Table and Chart"
_acmmm.org/ACMMM/2024/Conference — MM2024 Oral_

### Official Review · Reviewer_3ENm · 2024-05-21

**Rating:** 4
**Confidence:** 3

**Summary:**

This study focuses on the Multimodal Question Answering (MMQA) task, which entails answering questions based on diverse data representations such as tables, charts, and text. The authors claim that existing research has focused solely on two modalities, such as image-text QA, table-text QA, and chart-text QA, and lacks a comprehensive analysis involving all three modalities. To address this gap, the authors propose a Chinese multimodal reasoning-based Question Answering (QA) dataset, which encompasses three data modalities: text, tables, and charts. Additionally, this paper introduces a model named AED that consists of three agents to handle the different data modalities in the proposed dataset. The authors then conduct extensive experiments to validate the effectiveness of their proposed AED method.

**Strengths:**

1. This study introduces a Chinese multimodal reasoning-based Question Answering (QA) dataset, which encompasses three data modalities: text, tables, and charts. The authors conduct a comprehensive statistical analysis on the proposed dataset to elucidate its various properties and characteristics.
2. This paper is well organized and written.

**Limitations:**

1. The experiments in this paper are not comprehensive. The results presented in Table 4 do not provide a comprehensive evaluation of the authors' proposed AED method, as the authors have not compared it against a wide range of baseline approaches, such as GPT-4v and Llava. In contrast, the authors have utilized numerous other baseline methods in Table 5 to validate the effectiveness of their proposed AED approach. Consequently, the findings in Table 4 alone are insufficient to adequately support the efficacy of the authors' proposed AED method.

2. Concerns about evaluation metrics. The authors have only employed their newly proposed evaluation metric KM to conduct evaluation experiments. However, this evaluation metric lacks rationality and may require additional experiments or explanations to demonstrate its validity. Furthermore, the authors should report more evaluation results to validate the effectiveness of their proposed method, such as human evaluation.

**Suitability:**

2

---

### Official Review · Reviewer_jqcK · 2024-05-24

**Rating:** 4
**Confidence:** 4

**Summary:**

This paper introduces a Chinese reasoning-based QA dataset that includes an extensive collection of text, tables, and charts, and presents a multi-agent system implemented through collaborative deployment, information interaction, and collective decision-making among different agents.

**Strengths:**

1. Motivation and problem formulation is solid, with examples that help readers understand.
2. A multi-agent system AED for unified reasoning across text, tables, and charts modalities is proposed for MMQA research. The technical solution is feasible.

**Limitations:**

1.There are many domains and modes involved in the data set, and whether the amount of data is sufficient is not well proved.
2.The experiment is not sufficient, only the results of Chart QA related methods are shown, but not the other two modes, and the results of questions involving multiple modes (such as text and table) are not shown.

**Suitability:**

3

---

### Official Review · Reviewer_Mhso · 2024-05-25

**Rating:** 5
**Confidence:** 3

**Summary:**

This paper introduces CT2C-QA, a first-of-its-kind reasoning-based Chinese QA dataset that includes an extensive collection of text, tables, and charts, meticulously compiled from 200 selected webpages. Additionally, authors design a multi-agent system implemented through collaborative deployment, information interaction, and collective decision-making among different agents.

**Strengths:**

1. This paper constructs a first-of-its-kind reasoning-based Chinese QA dataset, which makes a contribution to this community.
2. This paper proposes an effective method on this benchmark, which even outperforms GPT4.

**Limitations:**

1. The experimental results lack human evaluation results.
2. How are these experts implemented? Are they gpt4?

**Suitability:**

3

---

### Meta-Review · Area_Chair_BFwy · 2024-07-06

**Recommendation:** Accept (Oral)
**Confidence:** 4

**Metareview:**

This paper introduces a Chinese multimodal reasoning-based QA dataset and a multi-agent system for unified reasoning across text, tables, and charts modalities. Reviewers praise the paper's contribution, solid motivation, and feasible technical solution but raise concerns about data sufficiency, experimental completeness, and evaluation metrics. Some reviewers suggest additional experiments, human evaluation, and comparison with more baseline approaches.

The authors have addressed some concerns in their rebuttal. With some revisions to address the remaining concerns, the paper can be considered to accept.